# In-situ photomechanical bending in a photosalient Zn-based coordination polymer probed by photocrystallography
Samim Khan [1], Shamim Ahmad[2], Sanobar Naaz[1], Niamh T. Hickey[3,4], Aditya Choudhury[5], Lauren E. Hatcher [6], Raghavender Medishetty [5] ✉, C. Malla Reddy[2,7] ✉, Sarah Guerin [3,4] ✉ & Mohammad Hedayetullah Mir [1] ✉

Photomechanical bending or mechanical flexibility in single crystals is an interesting landscape for innovative technological applications, including smart medical devices, molecular machines, artificial muscles, microrobots, and flexible electronic actuators. However, metal-organic crystals with multiple dynamic effects such as bending (in-situ), jumping, fracturing, and splitting in the absence of mechanical energy or temperature is interesting and relatively unexplored. The development of these materials presents significant challenges, requiring a thorough grasp of the underlying mechanisms for practical applications. Herein, we developed a Zn based 1D coordination polymer (CP) crystal $\{[Zn(DCTP)(4\text{-}nvp)_2]\cdot(CH_3OH)\}_n$ (**1**) $\{H_2DCTP = 2,5\text{-}dichloroterephthalic\ acid;\ 4\text{-}nvp = 4\text{-}(1\text{-}naphthylvinyl)pyridine\}$ which undergoes [2 + 2] cycloaddition under both UV irradiation and sunlight to generate a partially dimerized product of a two-dimensional coordination polymer (2D CP) $[Zn(DCTP)(rctt\text{-}4\text{-}pncb)]_n$ (**i$_{20}$1**). During UV irradiation, these single crystals exhibit photomechanical effects like jumping, bending, cracking, and swelling to relieve anisotropic strain from the light. Surprisingly, bent-shaped single crystals (**1b**) identical in structure to **1** were also obtained in-situ without any external stimuli, simply by keeping reaction mixture for an extended period. A time-resolved photocrystallographic study fully described the photoinduced structural transformation. Nanoindentation measurements complemented a DFT study of mechanical property trends for irradiated and bent Zn-based photosalient crystals.

In popular culture, the Mexican jumping bean exists as both a children's toy and curiosity owing to its seemingly random motion. It has been observed that the beans jump more as the temperature rises. The underlying reason is that the beans can be more energetic at higher temperatures and make sharp movements several times a minute while active. Thus, the locomotion of Mexican jumping beans opens up an idea to mimic such an effect in small crystals that undergo jumping, bursting, or swelling in response to external stimuli such as light[1–3]. Some crystalline materials are designed so that they exhibit multiple mechanical motions that include hopping, jumping, swimming, or even shattering into pieces when illuminated, which is termed the photosalient (PS) effect[4–9]. These materials, after light absorption, generate and accumulate strain due to structural anisotropy, which ultimately releases in the form of mechanical motion of the crystals[10]. Various parameters, such as cyclisation of diarylethylenes[6], isomerization[11], [4 + 4] photocycloaddition[12] reaction etc., have been reported for this kind of structural anisotropy. Recently, the impact of [2 + 2] cycloaddition on the PS effect has also reported by several groups, including our own[13–19]. In the case of [2 + 2] cycloaddition, the olefinic ligands undergo dimerization at a particular direction in the unit cell, leading to contraction of the cell from one side and expansion of the cell from other sides. If this anisotropy reaches a threshold level, the accumulated strain releases to optimize the energetics of the system, resulting in motion of the crystals observed macroscopically[20].

[1]Department of Chemistry, Aliah University, Kolkata, India. [2]Department of Chemical Sciences, Indian Institute of Science Education and Research (IISER) Kolkata, Nadia, West Bengal, India. [3]Department of Chemical Sciences, Bernal Institute, University of Limerick, Limerick, Ireland. [4]SSPC, The Research Ireland Research Centre for Pharmaceuticals, University of Limerick, Limerick, Ireland. [5]Department of Chemistry, IIT Bhilai, Raipur, Chhattisgarh, India. [6]School of Chemistry, Cardiff University, Cardiff, UK. [7]Department of Chemistry, Indian Institute of Technology Hyderabad, Hyderabad, India. ✉e-mail: raghavender@iitbhilai.ac.in; cmallareddy@gmail.com; Sarah.Guerin@ul.ie; chmmir@gmail.com

**Scheme 1 | Schematic representation of the photomechanical effects**. A synthetic route, different photomechanical behavior depending on morphology, nanoindentaion technique and finite difference calculations within the DFT framework.

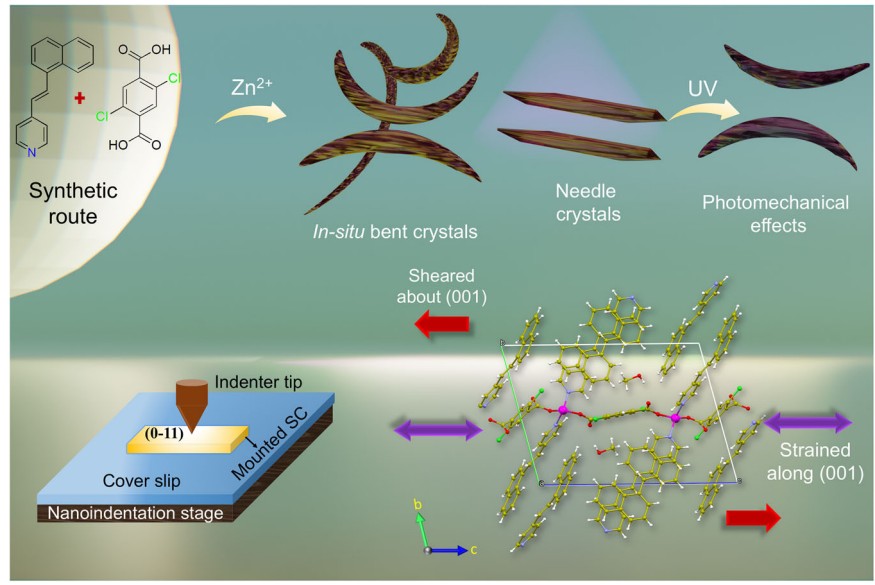

Designing such a system that exhibits photomechanical effects is itself a challenge. Generally, crystals violently explode during photoinduced transformation, and it often poses a serious challenge to obtain clean single-crystal-to-single-crystal (SCSC) transformations[21,22]. SCSC transformations provide exact structural insights into the transformed structures[23,24] with changes in molecular geometry and/or intermolecular interactions, and thus, it is the most valuable tool to investigate the mechanisms of photo-salient behavior. To achieve SCSC transformations, precautions are needed to preserve the crystallinity of the material. One possible way is to irradiate the crystal on the goniometer itself at low temperature. However, we have performed in-situ photocrystallography[25–27] at room temperature and obtained several dimerized structures. This gave us a chance to watch how the solid-state structural transformation progress in the crystals in real time, providing better insight into the photoconversion process of crystals.

Motivated by the potential applications of these dynamic photo-salient crystals, researchers are currently more focused on gaining a deeper understanding of the role of intermolecular interactions, such as strong hydrogen bonds and weak intermolecular interactions, and controlling design principles to tune their physicochemical properties[10,28–30]. Among various external stimuli, light-induced actuation presents significant advantages for macroscopic shape modulation over mechanical or thermally driven approaches, primarily due to its capacity for remote activation and highly controlled energy delivery. In this regard, in-situ photomechanical bending is even more intriguing since no input of external energy or UV light is needed[31–34]. In spite of considerable research in this field, two prominent obstacles remain to be tackled; low efficiency and lack of multifunctionality—highly desirable but challenging objectives to achieve. The actuation efficiency is unpredictable and is practically challenging to assess for small objects such as metal-organic crystals. At the same time, achieving multiple photomechanical effects from the same compound is also challenging due to the difficulties in predicting the crystal packing, intermolecular interactions, and crystal behavior.

Herein, we report 1D CP crystals of formula {[Zn(DCTP)(4-nvp)₂]·(CH₃OH)}ₙ (**1**) that undergo topochemical [2 + 2] cycloaddition under UV irradiation as well as sunlight to generate a partially dimerized product of 2D CP [Zn(DCTP)(*rctt*-4-pncb)]ₙ (**i₂₀1**). During this photo-reaction, the Zn crystals show multiple photomechanical effects such as jumping, bending, swelling, and splitting. The needle-shaped crystal undergoes bending, flipping, and swelling. This form also grows in-situ, which may be due to the slight absorption of visible light. In addition, bent-shaped single crystals (**1b**), structurally identical to **1**, were also

unexpectedly formed in-situ without any external stimuli, simply by allowing the reaction mixture to sit for an extended period of 2 weeks. We have undertaken a photocrystallographic study to get insights into these mechanical motions during structural transformations. Furthermore, structure-property relationships of irradiated and bent Zn-based photo-salient crystals have been established via a DFT study and complimentary nanoindentation measurements. Here, we predominantly use density functional theory (DFT) to predict the static and dynamic mechanical properties of each crystal to understand the atomic-scale mechanisms and mechanical shifts that occur under irradiation. These DFT calculations serve to understand the relative internal stress build-up in an irradiated photo-salient crystal. Nanoindentation measurements, a reliable technique to quantify the mechano-structure-property relationship of the crystals, have been performed, and the results are correlated to DFT-predicted values (Scheme 1).

## Results and discussion
### Synthesis and characterization of 1, i₅1, i₁₀1, i₂₀1, and 1b

The compound {[Zn(DCTP)(4-nvp)₂]·(CH₃OH)}ₙ (**1**) was synthesized by a slow diffusion method via layering of H₂DCTP and 4-nvp ligands onto a solution of Zn(NO₃)₂·6H₂O in the presence of Et₃N (Supplementary Scheme 1). Single crystal X-ray diffraction (SCXRD) analysis reveals that compound **1** crystallizes in the triclinic crystal system with space group P$\bar{1}$ and Z = 2. The asymmetric unit consists of one Zn(II) atom, two 4-nvp, and one DCTP ligand (Supplementary Fig. S1). The asymmetric unit also contains a disordered methanol molecule which forms a strong hydrogen bond with open oxygen atoms from bridging DCTP ligands. Each Zn(II) center has tetrahedral geometry and is ligated to two DCTP and two 4-nvp ligands. The bridging DCTP ligand connects two Zn(II) centers to generate a 1D CP, and 4-nvp are exposed at both sides of the chain. Interestingly, in the solid-state structure, one of the two 4-nvp ligands is aligned with a 4-nvp ligand of the adjacent chain and produces a head-to-tail arrangement with a distance between the centers of two C=C bonds of 3.81 Å (Fig. 1a), which is highly favorable for photochemical cycloaddition following Schmidt's criteria (<4.2 Å)[35]. However, the compound **1** undergoes partial dimerization, and the photoproducts i₅1, i₁₀1, and i₂₀1 are obtained via photocrystallographic studies by irradiating crystal **1** for 5 min, 10 min, and 20 min (Supplementary Fig. S2) as discussed below. In addition, in situ bent crystals (**1b**) are obtained by keeping the reaction mixture for 2 weeks. PXRD patterns of before and after UV irradiation are shown in Fig. 1d, whereas needle and bent crystal phase purity are depicted in Supplementary Fig. S3–S4.

**Fig. 1 | Crystal structure of the synthesized compounds. a** A perspective view of the compound **1**. **b** Partially dimerized compound **i₂₀1**. **c** In-situ bent compound **1b**. **d** PXRD patterns of simulated (black) and as-synthesized before UV (red) and after UV irradiation (blue) of compound **1**. **e** Optical microscopic image of needle crystal **1** before irradiation. **f** Bending crystal of **1** upon UV irradiation. **g** In-situ obtained bent crystal **1b**.

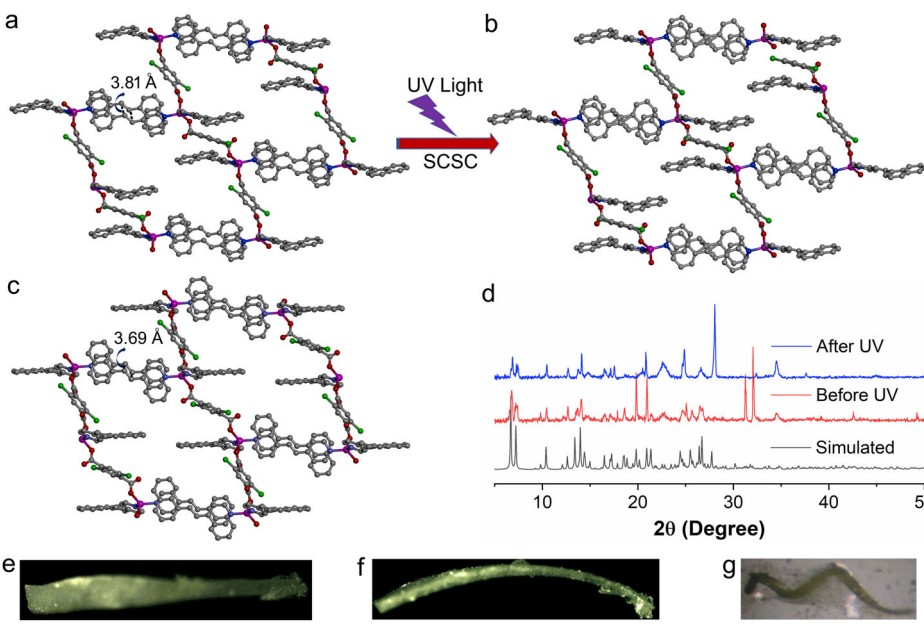

## Photocrystallographic studies on [2 + 2] photocycloaddition reactions

It is clear from the crystal packing and alignment of olefin pairs that compound **1** is supposed to undergo a quantitative photochemical [2 + 2] cycloaddition reaction to generate a 2D CP. Thus, the photoreactivity of crystals **1** was investigated by photocrystallographic techniques. Photocrystallography is a technique used to study the structural changes in crystals when they are exposed to light. Herein, [2 + 2] cycloaddition reaction is investigated by a photocrystallographic study. The crystals are exposed to UV light in situ on the diffractometer, and the SCXRD data are collected at various stages of the reaction during a particular time interval. In this regard, we have irradiated one suitable single crystal (0.5 mm long) with a slightly lower energy wavelength with a 390 nm LED. The selected wavelength was chosen based on the common solid-state approach of using light within the tail of the absorption band. This is a compromise, as it still provides enough energy for the photoexcitation process, but it is not absorbed very strongly at the surface of the crystal, therefore, this helps the light to penetrate further through the crystal bulk and maximize the photoexcited state population[36,37]. Irradiation at 390 nm resulted in 12.4(1)% conversion to the photodimerized product after 5 min (**i₅1**), 15.1(2)% conversion after 10 min (**i₁₀1**) and 16.4(1)% conversion after 20 min of irradiation (**i₂₀1**) (Fig. 1b). However, further irradiation causes the crystals to shatter and thus, it seems that 17% is the maximum conversion where the crystals can be sustained before breaking. As expected, the photoirradiated crystal generated relatively marginal overall data with extensive disorder due to the fast photodynamic reaction with the major packing change.

## Isolation of in-situ bent crystals

Photomechanical bending is quite interesting in the case of photosalient effect and is often triggered by UV light. However, in-situ bending of a metal-organic crystal is rare. Interestingly, we obtained bent crystals in a similar condition to **1**, except for a longer time of 2 weeks, but the single crystal structure (**1b**) suggests 9% of cycloaddition and thus that the crystals may have been exposed to visible light (Fig. 1c). These bent crystals sustained crystallinity during SCXRD data collection at 100 K. However, highly bent crystals (bent position) diffracted very poorly with diffused reflections at the bent region (supplementary Fig. S5). In the case of the bent crystal, the distance between neighboring C=C moieties decreased to 3.69 Å along with the formation of partially dimerized cyclobutane (supplementary Fig. S6), clearly indicating that the PS effect occurs via cracking, swelling, bending, followed by jumping and bursting. Optical images of crystals of **1** and in-situ bent (**1b**) crystals are also shown (supplementary Fig. S7).

## Underlying photoreactivity and photomechanical effects

Furthermore, the photoreactivity of compound **1** was investigated by ¹H NMR spectroscopy, which shows the appearance of the peaks due to cyclobutane protons at ca. 5.2 ppm (supplementary Fig. S8 and S9). The conversion rate of the [2 + 2] photodimerization and completion of the reaction were verified by time-dependent ¹H NMR graph (supplementary Fig. S10 and S11). Besides, the progress of photo conversion of needle and bent crystals of **1** over various time intervals was depicted in Supplementary Fig. S12 and S13. These results show that compound **1** undergoes photo-dimerization to produce a 2D CP. However, in order to authenticate the photodimerization of **1**, a SCSC transformation was attempted by irradiating the crystals of **1** with relatively harder UV light (wavelength 365 nm), which caused the crystals to shatter on the diffractometer. These results prompted us to check further the PS effect of the crystals. To this end, we recorded a video of the PS effects of the crystals using a microscope equipped with a high-resolution camera in presence of lower energy UV light (~390 nm). A range of intriguing photomechanical effects, including cracking, bending, flipping, swelling, and jumping, were observed for the needle crystals over 20 min of UV irradiation (refer to Supplementary Videos SV1–SV4). Herein, crystals relax by mechanical reconfiguration (reshaping) without fragmentation and remain in the state of mechanical equilibrium during the whole process. The bending direction, away from the light source, can be attributed to the expansion of the (partially) reacted crystal surface. Additionally, the PS effect can be visualized using field emission scanning electron microscopy (FESEM) images of the crystals both before and after photoirradiation (Fig. 2).

In order to analyze why the PS effect is observed in **1**, we report the variations in cell parameters before and after exposure to photoirradiation. Determining the structure of PS crystals poses significant challenges, as the PS effect frequently results in only fragmented debris rather than complete crystals. The primary methods for acquiring the crystal structure involve either recrystallizing the photoproduct or achieving SCSC transformation, both of which are exceedingly difficult. To our knowledge, only a few documented instances of successfully determining the structure of a dimerized PS crystal through photochemical [2 + 2] cycloaddition via SCSC transformation have been reported. Although we could not obtain the structure of quantitative dimerization, we could attain the partially dimerized (~17%) structure of photoproduct **i₂₀1**. Partial dimerization results in alterations in both the cell axes and the cell volume (Table S1). The cell parameters reveal that the cell volume increases by 8.43 Å³ (0.46%) from **1** to **i₅1** (Supplementary information Table S1). Extensive reports

**Fig. 2 | Illustration of PS effect. a** FESEM images of crystal **1** before UV irradiation. **b** FESEM images of in-situ crystal **1b** after popping under UV irradiation. **c** FESEM images of crystal **1** after popping under UV irradiation. **d** FESEM images of crystal **1** after popping under UV irradiation in small scale.

show that the PS effect resulting from photodimerization is observed when there are notable changes in cell parameters. In scenarios where photo-dimerization causes a change in cell volume—whether it decreases or increases—the crystals can experience excessive stress, which is then relieved through mechanical motion.

## DFT-calculated mechanical properties

In order to investigate the anisotropic mechanical properties of PS crystals, DFT calculations were carried out to predict the elastic stiffness tensor of each crystal[38]. Figure 3 shows the computed stress-strain curves of the crystal at different intervals of irradiation. Before and after irradiation, the crystal has clearly defined elastic and plastic regions, along with fracture points at 20% and 18% strain, respectively. However, the induced stress needed for the pre-irradiated crystal to break is nearly double that of the post-irradiated crystal. Post-irradiation, the crystal structure can withstand higher strain with a severely reduced induced stress. While all other crystal structures follow very similar trends (both shape and magnitude) in the elastic region, the post-irradiation structure is predicted to demonstrate completely distinct and stable mechanical behavior.

Figure 3 shows the computed stress-strain curves at different points of irradiation; pre-irradiation, 5 min in, 10 min in, and 20 min in. These stress-strain curves show the crystals to have continuous plastic behavior from the linear region up to 20%; however, it must be noted that none of the irradiated structures could converge at 25% strain, indicating mechanical instabilities as the induced stress increases near-linearly. The ultimate tensile stress at 5 min is slightly higher than at 10 or 20 min. The stress-strain curves at 10- and 20-min show near-identical behavior. As mentioned previously, the

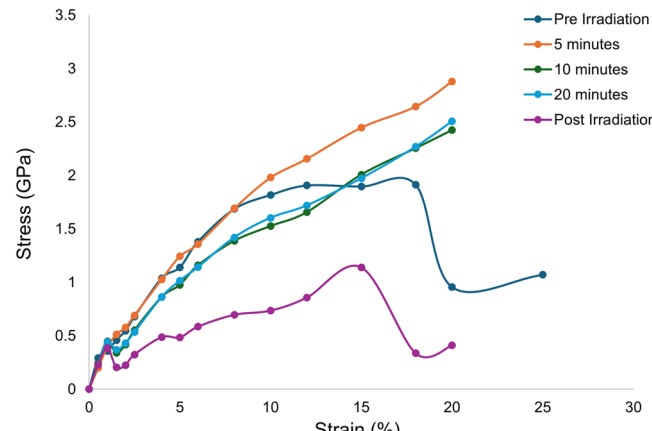

**Fig. 3 | Stress-strain response of photosalient crystal under UV irradiation.** Comparison of the computed stress-strain curves of a photosalient crystal along the crystallographic c axis prior to irradiation (navy), 5 min after irradiation (orange), 10 min after irradiation (green), 20 min after irradiation (blue), and after irradiation and bending (purple).

three crystals undergoing irradiation have similar elastic behavior to the pre-irradiated structure. However, the crystals deviate in behavior circa 10% strain. Pre-irradiation the crystal structure has the features of a "conventional" stress-strain curve, demonstrating elastic, plastic, and necking

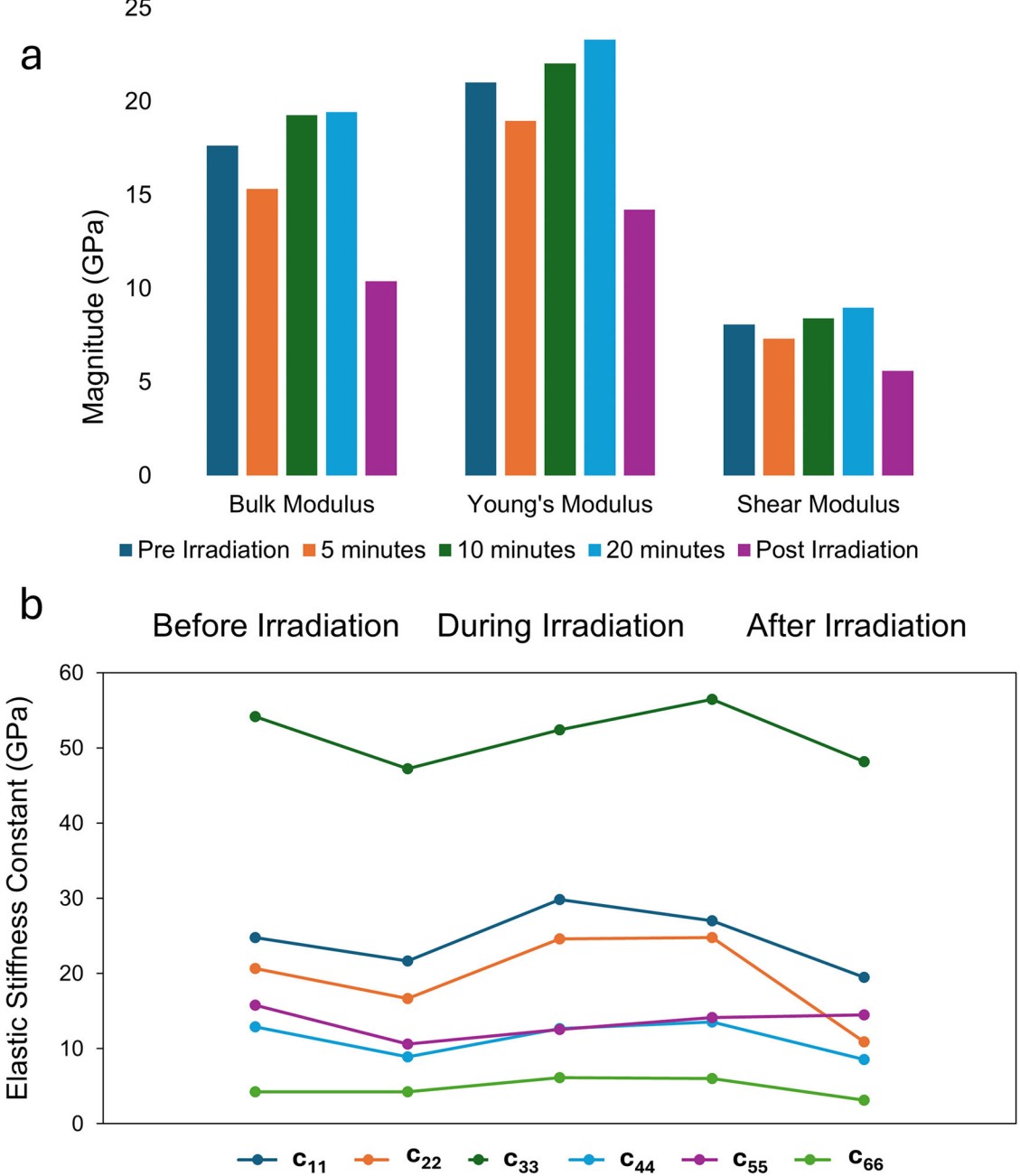

**Fig. 4 | Tracking Mechanical Properties of Photosalient Crystals using DFT. a** Comparison of the Bulk, Young's and Shear Moduli at different points of irradiation. **b** Comparison of 6 different elastic stiffness constants at different points of irradiation.

behavior before fracturing. During irradiation, the structure shows no discernible necking or fracture point, with stress increasing proportionally to strain. The DFT-optimized ground-state structures at each irradiation point are shown in supplementary Fig. S17, showing no distinct changes in the unit cell outside of the cycloaddition process and the structural changes discussed previously (Tables S1–S2).

Figure 4a shows the derived Bulk, Young's, and Shear Moduli of the five crystal structures at different points of irradiation. The trend in the three moduli follows the same pattern as the longitudinal directional elastic stiffness values- starting high at the pre-irradiation mark, falling at the 5-min mark, rising at the 10- and 20-min marks, and then falling again at the post-irradiation mark. The predicted Shear Modulus is of similar magnitude to the experimentally measured mechanical response, as discussed below, as the axis of indentation is not along a crystallographic axis. We have previously observed the nanoindentation-

measured stiffness matching the DFT-predicted shear modulus in instances where the probe approaches the crystallographic axis at an angle[39]. Overall, irradiating the crystal is predicted to decrease the Bulk, Young's, and Shear Moduli, indicating reduced brittleness and increased flexibility. This trend is seen in the elastic stiffness constants $c_{11}$, $c_{22}$, $c_{33}$, and $c_{44}$, but not in the shear constants $c_{55}$ or $c_{66}$ (Fig. 4b). These diagonal stiffness matrix components ($c_{ij}$, where $i = j =$ the direction of stress and strain in the crystal; 1,2,3 = longitudinal, 4,5,6 = shear) give information on the induced stress in the same direction as the applied strain and corresponds to the linear region of the stress-strain curve in that direction. Therefore, their magnitude broadly follows the same principles as Young's Moduli- with higher values corresponding to more rigid materials, and lower values corresponding to more flexible/compliant materials. The $c_{55}$ constant has similar values at pre and post irradiation (15.8 GPa vs 14.5 GPa). The $c_{66}$ constant is predicted to rise during

irradiation and fall post-irradiation. This is consistent with the high predicted $c_{33}$ values of up to 56 GPa, indicating that the crystallographic $c$ axis has the highest mechanical rigidity, and a notably high directional stiffnesses amongst molecular crystals as a material class[40]. The full $6 \times 6$ calculated elastic stiffness tensors for all crystal forms are in Supplementary Tables S3–S7, and the 2D and 3D visualizations of the elastic anisotropy are shown in Supplementary Figs S18–S20.

### Determination of mechanical properties by nanoindentation

Despite many studies on the nano-mechanical properties of molecular crystals[41–45], fewer studies exist in the domain of single crystalline frameworks or co-ordination polymers[46–48]. The pristine crystal (before irradiation) features a one-dimensional polymeric chain (Fig. 5a) extending along the unit cell's "c" axis, stabilized by hydrogen bonding with neighboring methanol. In order to shed light on the predicted mechanical behaviors, we have performed nanoindentation measurements on the major face (0–11) (which is the most prominent face) of the pristine crystals (Fig. 5a) at a peak load of 5 mN. Schematic crystal faces and respective packing diagrams of crystal **1** and **1b** are depicted in Supplementary Figs. S14–15, respectively. However, nanoindentation could not be performed on the irradiated or bent crystals due to inhomogeneity in the surface morphology and loss of single crystallinity. Nanomechanical quantification gave us a concrete foundation to verify the mechano-structure-property relationship[49,50]. Characteristic load−displacement ($P-h$) curves (Figs. 5b and S15a) were found to be smooth, which signifies the absence of sudden notable displacement bursts (pop-ins)[51]. A post indentation 2D scanning probe microscopy (SPM) image (Supplementary Fig. S16c) and 3D image (Fig. 5c) of the nanoindentation impression of crystal **1** were obtained. The average elastic modulus ($E$) ($6.32 \pm 1.14$ GPa) and hardness ($H$)

($424.13 \pm 43.4$ MPa) values suggest the crystals are moderately robust, which were obtained by standard Oliver−Pharr (O − P) method[52,53]. The moderate values for the mechanical property parameters (elastic modulus and the hardness) can be attributed to the low density of strong intermolecular interaction and distal π−π interactions, particularly along the axis of measurement. The large difference (Table 1) between the average maximum depth (hmax; $825.87 \pm 49.41$ nm) and final depth (hf; $456.99 \pm 38.54$ nm) suggests a post-indentation colossal elastic recovery. The distribution of the $E$ and $H$ values is shown in histogram plots (Fig. 5e, f), whereas the height profile from the indentation impression is shown in Fig. 5d.

### Conclusions

In summary, we have synthesized a Zn(II) 1D CP which undergoes photochemical [2 + 2] cycloaddition to generate a 2D CP accompanied by cracking, bending, swelling, flipping, and jumping of the crystals. Most notably, in-situ bending of crystals was obtained by keeping the reaction mixture for a longer period (2-3 weeks). The photodimerized intermediate structures were obtained via time-resolved photocrystallography that allowed us to gain insights into the structural transformation and mechanism behind the photosalient effect. The PS effect here is a response for seeking mechanical stability and higher symmetry, where UV radiation creates a complex network of internal restrictions and relaxations that can only be resolved through mechanical motion. The results presented here suggest that utilization of dynamic crystals for simple photochemical reaction under UV or visible light (green process) can turn light into useful work and pave the way for use of these materials as lightweight, efficient micromachines. We have performed DFT calculations and nanoindentation measurements, which suggest that the structure, morphology, and size can be important tools for the type and magnitude of the related mechanical effects.

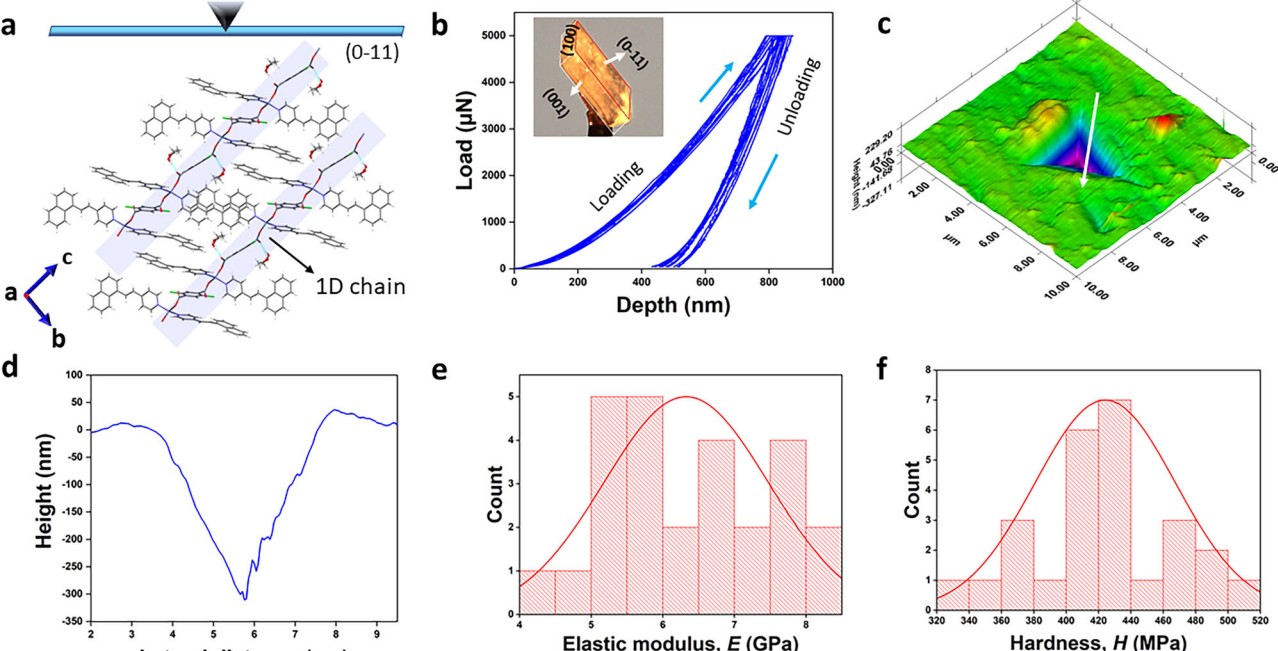

**Fig. 5 | Experimental nanoindentation benchmarks of SC mechanical properties.** Nanoindentation studies. **a** Crystal packing with 1D chain along "c" axis and indentation direction perpendicular to the (0-11) plane. **b** multiple load ($P$) – displacement ($h$) curves (indexed faces of the crystal in the inset). **c** 3D indentation impression. **d** Height profile from the indentation cater (direction is shown in white arrow in the 3D impression). **e, f** Histogram plots representing the distribution for elastic modulus and the hardness.

**Table 1 | Table for the parameters from the nanoindentation measurements**

| Elastic modulus ($E$, GPa) | Hardness ($H$, MPa) | Maximum depth ($h_{max}$, nm) | Final depth ($h_f$, nm) |
|---|---|---|---|
| $6.32 \pm 1.14$ | $424.13 \pm 43.4$ | $825.87 \pm 49.41$ | $456.99 \pm 38.54$ |

## Materials and methods

All chemicals purchased were reagent grade and were used without further purification. Elemental analysis (carbon, hydrogen, and nitrogen) was performed on a Perkin–Elmer 240C elemental analyzer. Infrared spectrum in KBr (4500–500 cm$^{-1}$) was recorded using a Perkin–Elmer FT-IR spectrum RX1 spectrometer. $^1$H NMR spectra were recorded on a 400 MHz Bruker Avance 400 FT NMR spectrometer with TMS as an internal reference in DMSO-$d6$ solution. The solid-state photoluminescence measurements were made using a Horiba Fluorolog with a solid-state sample holder. The solid-state photoluminescence measurements were made using a Horiba Jobin Yvon Fluoromax-4 Spectrophotometer (Excitation wavelength–390 nm) with a solid-state sample holder. A JEOL JSM-6701F Field Emission Scanning Electron Microscope (FESEM) was used to obtain SEM images. The photodimerization reaction was carried out using a Luzchem photoreactor (8 W UVA lamps) at ~350 nm and room temperature. Crystalline ground powder was packed between the glass slides and irradiated under UV light. Glass slides were flipped at regular intervals of time to maintain uniform exposure of UV radiation. Photosalient effects were studied by irradiating good quality single crystal with UV light from a Xe source using a MAX-350 optical photoreactor under a microscope equipped with a high-quality camera for capturing videos.

## Syntheses of compounds

**Synthesis of 1:** A solution of 4-nvp (0.046 g, 0.2 mmol) in MeOH (2 mL) was slowly and carefully layered onto a solution of Zn(NO)$_3$·6H$_2$O (0.059 g, 0.2 mmol), in H$_2$O (2 mL) using a 2 mL 1 : 1 ( = v/v) buffer solution of MeOH and H$_2$O followed by layering of H$_2$DCTP (0.047 g, 0.2 mmol) neutralized with Et$_3$N (0.021 g, 0.2 mmol) in 2 mL EtOH. The reaction mixture was kept in the dark. The light-yellow color needle-shaped crystals of [Zn(4-nvp)$_2$(DCTP)]·CH$_3$OH (**1**), were obtained after four days (0.098 g, yield 62%). Elemental analysis (%) calculated for C$_{43}$H$_{32}$Cl$_2$N$_2$O$_5$Zn: C, 65.13; H, 4.07; N, 3.53; found: C 65.01, H 4.2, N 3.42.

**Synthesis of i$_5$1, i$_{10}$1 and i$_{20}$1**. The compounds **i$_5$1, i$_{10}$1 and i$_{20}$1** were obtained by in-situ UV irradiation of single crystals of **1** using a LED lamp centred at ~390 nm wavelength for 5, 10, and 20 min to obtain the photodimerized (intermediate) product in almost quantitative yield.

**Synthesis of 1b**. The compound **1b** was obtained in almost similar process as of **1**, only the solution was kept for 2–3 weeks.

## X-ray crystallography

Single crystals of all compounds having suitable dimension were used for data collection using a Rigaku Gemini A-Ultra diffractometer equipped with graphite-monochromated MoK$_\alpha$ radiation ($\lambda = 0.71073$ Å) and an Atlas CCD detector for **1**, and a Rigaku (XtaLAB Synergy custom system with HyPix detector) diffractometer with mirror monochromated CuKα ($\lambda = 1.54184$ Å) source for **1b**. Data reduction was carried with the Rigaku CrysAlisPro software[54]. The crystal structure was solved by Intrinsic Phasing using the ShelXT program[55]. The structure was refined by the full-matrix least-squares method using ShelXL 2018/3[56], via the Olex2 interface[57]. Non-hydrogen atoms were refined by the help of anisotropic displacement parameters. All the hydrogen atoms were placed in their geometrically perfect positions and constrained to ride on their parent atoms. Crystallographic data for all compounds are summarized in Supplementary Table S1, and selected bond lengths and bond angles are given in Supplementary Table S2.

## In-situ photocrystallography

In-situ irradiation of the crystals of the diffractometer was achieved using a bespoke LED ring array set-up as described in a previous publication[58]. The array positioned 6 × 3 mm LEDs ($\lambda_p = 390$ nm, UV3TZ-390-30, 40 mW, 30° viewing angle) approximately 1 cm away from the crystal in a uniform arc, and the crystal was additionally rotated about the diffractometer $\varphi$-axis during the irradiation period to ensure even illumination on all faces. After each selected irradiation period, the LEDs were turned off and a standard

SCXRD data collection was performed in the dark, as outlined above, to obtain the crystal structure of the photoexcited state. Photocrystallographic data collection and data processing information are detailed in Supplementary Method 1.

## Computational methods

Unit cells were strained in the range of 0–25% by multiplying the equilibrium unit cell matrix by a simple strain matrix that increased the dimensions of the cell along the axis of the net unit cell dipole at equilibrium to replicate the observed experimental bending in the (001) plane. The mechanical properties were predicted from periodic DFT calculations[59] using CP2K[60] and were obtained following Vanderbilt's work on the systematic treatment of perturbations[61]. More precisely, the relaxed-ions approach described by Eq. 1 was used to avoid the computation of the pseudo-inverse of the Hessian matrix, and obtain the elastic tensors with a single derivation:

$$C_{ij} = -\frac{\partial \sigma_i}{\partial \eta_j} \tag{1}$$

A perturbative approach is used to perform the derivation. The system is strained along the 6 Voigt directions with a negative and positive perturbation. The stress tensors are then used to build the aforementioned tensors using finite differences. The unique rotations of the native space group are then applied to the tensors to symmetrize them. The Bulk, Young's, and Shear moduli are presented as Voigt-Reuss-Hill averages[62], using similar derivation methods to the ELATE program[63], and crystal structures were visualized using VESTA[64].

## Nanoindentation details

Suitable single crystals were mounted using feviquick (cyanoacrylate) glue on a stainless-steel round shaped sample holder having smooth surface. The experiment was carried out using a nanoindenter (Hysitron Triboindenter, TI Premier, Minneapolis, USA) with a three-sided pyramidal Berkovich diamond indenter tip of radius 120 nm having an in situ scanning probe microscopy (SPM) facility. Before nanoindentation testing, the tip area function was calculated from a series of indentations on a standard fused quartz sample. The maximum load was used as 5 mN with 5 s duration for loading-unloading segments, and a 2 s holding period was applied (Supplementary Fig. S16). Several indentations were performed for each crystal, and SPM images of the indentation impressions were captured immediately just after unloading. The obtained $P-h$ curves were analyzed using the standard Oliver−Pharr method[52,53] to extract the required parameters such as elastic modulus ($E$) and hardness ($H$) of the crystals.

## Data availability

The authors declare that all data supporting the findings of this study are available within the Article and its Supplementary Information files and from the corresponding author upon request. The crystal structures of **1, i$_5$1, i$_{10}$1, i$_{20}$1, and 1b** are available free of charge from the Cambridge Crystallographic Data Centre under reference numbers 2385118 (**1**, Supplementary Data 1), 2385119 (**i$_5$1**, Supplementary Data 2), 2385120 (**i$_{10}$1**, Supplementary Data 3), 2385121 (**i$_{20}$1**, Supplementary Data 4) and 2385122 (**1b**, Supplementary Data 5).

## Code availability

No custom code or mathematical algorithms were required for the central conclusions of this manuscript. Mechanical properties can be calculated using https://www.cp2k.org/as per the methods section, and https://progs.coudert.name/elate.

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

## Acknowledgements

M.H.M. is thankful to SERB India for a Core Research Grant (Grant No. CRG/2022/001842, dated 19/12/2022). R.M. thanks DST-SERB for SRG/2019/001508 and IIT Bhilai for RIG/2003600. S.G. and N.H. would like to acknowledge funding from Science Foundation Ireland under grant numbers 21/PATH-S/9737 and 12/RC/2275_P2 respectively, as well as ongoing support from the Irish Centre for High-End Computing (ICHEC). S.G. is funded by the European Union under ERC Starting Grant no. 101039636.

## Author contributions

M.H.M., R.M., and C.M.R., and S.G. designed the project. S.K. and S.N. synthesized the crystals, L.H., S.K., and S.A. carried out the X-ray structural data analysis, S.K. performed NMR, PXRD measurements S.K. and A.C. captured videos of photosalient effect. S.A. performed Nanoindentation study. A.C. and S.K. collected SEM data. N.H. and S.G. performed the computer simulations, M.H.M., R.M., C.M.R., and S.G. directed and supervised the overall project. All authors contributed to writing the manuscript.

## Competing interests

The authors declare no competing interests.
