## [Transparent Peer Review file · Communications Chemistry]

In-situ photomechanical bending in a photosalient Zn-based coordination polymer probed by photocrystallography

Corresponding Author: Dr Mohammad Hedayetullah Mir

Version 0:

Reviewer comments:

Reviewer #1

(Remarks to the Author)

S. Khan et al. have reported the photosalient effect in a one-dimensional zinc coordination polymer. Indeed, photosalient phenomena driven by [2+2] cycloaddition in coordination polymers are rare due to the uncertainty of their aggregated structures. However, this manuscript contains several points that need revision, and therefore, a major revision is recommended.

Points for Revision

- Figure 1 is too small and difficult to see. A clearer figure should be provided.
- A figure illustrating at least the parallel alignment of C=C bonds should be included.
- A graph showing the time-dependent conversion rate of the [2+2] photodimerization is necessary. It should clarify how long the irradiation takes for the conversion to reach saturation.
- The paper describes single-crystal structural analysis after 390 nm light irradiation, but the method used to determine the conversion rate is unclear. Since ¹H NMR is typically used to determine conversion rates, it is important to explain how it was evaluated in this study.
- Photos of the sample before and after the photosalient effect should be included in the main text.
- Even if determining the structure of the complex after [2+2] photodimerization is challenging, the structure of the ligand should be accessible. Could the complex be decomposed with hydrochloric acid to extract and analyze the organic product of the [2+2] reaction?
- Figures 3 and 4 appear to show the same data. There seems to be no need to separate them; only Fig. 4 should be retained.
- The computational results should support the experimental data. Consider whether they can be discussed in relation to the nanoindentation results.
- The meaning of the values for c11, c22, c33, c44, c55, and c66 in Figure 5 is unclear. An explanation should be provided.
- Figure S6 lacks a scale bar.
- The differences between Figure S8a-e are unclear, and further explanation is required.
- Examples of bending behavior induced by [2+2] cycloaddition are rare. For instance, please add the following references: Chem. Eur. J., 2024, e202401564; Chem. Mater., 2024, 36, 8338–8348; J. Mater. Chem. C, 2021, 9, 16762–16770; J. Am. Chem. Soc., 2023, 145, 6024–6028; CrystEngComm, 2025, 27, 127–137.

Reviewer #2

(Remarks to the Author)

The manuscript by Khan et al. presents findings on the photosalient effects observed in a one-dimensional coordination polymer of zinc(II), an unusual phenomenon for metal-containing crystalline materials. The authors characterized the material using time-resolved photocrystallography and nanoindentation, supplementing their investigation with DFT studies. While the topic is of significant interest to a broad audience, making it suitable for publication in the journal Communications Chemistry, the manuscript requires substantial improvements before it can be considered for acceptance.

1. The authors claim that there are two polymorphic forms of the zinc-based 1-D polymer. If this is indeed the case, the preparation details for both forms should be clearly articulated, emphasizing the differences in their preparation methods. The information provided regarding the preparation of form II is insufficiently detailed; specifically, it is unclear whether both forms were stored in the dark, how long the crystals remained in their mother liquors, and what procedures were followed after the crystals were isolated—were they also kept in the dark, for instance? Additionally, it would be ideal to demonstrate phase purity by presenting PXRD patterns, along with their comparisons to the respective patterns calculated from the crystal structures and/or their mutual overlaps.

Furthermore, a systematic labeling of the two forms in the manuscript would be necessary.

2. In the abstract, the authors claim that during 'photoinduced transformation, the crystals exhibited diverse photomechanical movements, such as jumping, bending, flipping, splitting, and swelling in response to UV light.' However, only bending and splitting/explosions were described in the MS.

3. The nanoindentation was performed on (0-11) crystal face, but it is unclear why this specific face was selected for analysis. What other faces were developed on the crystals of the two forms? Additionally, what is the relationship between the crystal structures and the crystal morphology (i.e., crystal faces) of these two forms? Similarly, regarding the DFT studies, it is not clear why the authors chose to focus on the strain induced in the c direction; it should be mentioned. Please pay attention to the distinctions in labeling crystallographic directions and crystal faces.

4. The differences in moduli (Shear, Bulk, Young's) before, during and after bending in relation to 'elastic stiffness constants' (c_{11} , c_{22} ..., c_{66}) were discussed, and the behaviors were further related with the reduced/increased brittleness/flexibility. It would be beneficial to provide a more detailed explanation (in ESI) why the authors selected these specific matrix elements and how their changes relate to the brittleness and flexibility of crystals. Additionally, the authors mentioned visualizing the calculated stiffness constants by the ELATE/VESTA program including 3D presentations and/or 2D sections in the main text or ESI would be advantageous.

5. While Scheme 1 is a valuable addition, a more comprehensive scheme that includes additional details would be more appropriate. Such a scheme would aid readers in understanding the complex behavior of the CP-based material.

6. Page 11, In 228; refs [42-44]; please check the reliability of the citation.

Reviewer #3

(Remarks to the Author)

In my view this work is way too premature for publication. The results may be of potential interest to the crystal engineering community, but (1) the manuscript is poorly written, and (2) insufficient supporting evidence is provided for the discussion and conclusions. The presentation of the work is sloppy and the results don't stand out as being unusual for crystals that undergo [2+2] cycloaddition reactions. How reproducible are the nanoindentation results?

In addition to the above, the following needs to be addressed if the authors are planning to submit this work again:

The formula $[Zn(DCTP)_2(rctt-4-pncb)]_n$ is incorrect everywhere in the manuscript. It should be $[Zn(DCTP)(rctt-4-pncb)]_n$ {note, there is only one DCTP per Zn ion}

Inclusion of a chemical scheme showing the formula unit would be useful to the reader.

Line 47: These materials, after light adsorption, generate and accumulate strain ...
"adsorption" should be "absorption"

Line 49: "di-arylethylenes" should be "diarylethylenes"

Line 50: insert "of" between "kind" and "structural"

Line 55: "energetic" should be "energetics"

Line 55: "resulting in motion" - insert "in"

Line 55/56: change to "observed macroscopically"

Line 57: "crystals violently explode and shatter into pieces" - things that explode generally shatter into pieces, so this is a tautology

Line 61: change "tools" to "tool"

Line 71: "Of the external stimuli, actuation by light comes with advantages for control over the macroscopic shape in comparison to mechanical or temperature-driven most notably, the possibility for remote control and precise input of energy" very poor sentence structure.

Line 73: "In this regard, in-situ photomechanical bending is even more intriguing since no input of external energy or harmful UV light is needed". If bending is due to irradiation, then how can there be no input of external energy?? The inclusion of "harmful" seems spurious because this doesn't need to be an issue for the manipulation of materials.

Line 81: "formulae" should be "formula"

Line 85: "Unexpectedly, bent shaped single crystals (1b) were also obtained in-situ without applying any external stimuli".

Firstly, it's not clear what "in situ" is referring to. Do the authors mean in situ in the vial, or in situ in the X-ray beam? Either way, there are external stimuli that cannot be ruled out, including temperature, ambient light and (possible) X-ray photons.

Line 86: "We also observed two polymorphs with different photomechanical properties." On what basis have the authors decided that 1 and 1b are polymorphs. This is never really discussed. Besides, the unit cell parameters, are similar and the space groups are the same. The packing arrangements are also sufficiently similar to suggest that these are not polymorphs.

Line 87: "wherein crystal" - insert "the"

Line 89: "flipping and swelling and this form also grow in-situ which may be due to the slight absorption of visible light". Change "grow" to "grows". Again, what does "in situ" refer to? The vial or the X-ray beam?

Line 107: "forms a strong hydrogen bond" - insert "a" as shown

Line 109: "generate a 1D CP"

Line 110: "exposed at both sides of the chain"

Line 116: " In addition, in situ bent crystals (1b) are obtained by keeping the reaction mixture for longer time." Omit "time"; again, what does in situ mean?

Line 132: "A photocrystallographic study of a [2+2] cycloaddition reaction involves using photocrystallography to investigate the [2+2] cycloaddition reaction". This sentence contains no useful information, and states the obvious twice.

Line 127: "The reason behind choosing the certain wavelength, in the solid-state, it is typical to choose a wavelength in the tail of the absorption band." Very poor sentence construction

Line 132: "On irradiation at 390 nm, it resulted in 12.4% conversion to the photodimerized product after 5 min (i51), 15.1% conversion after 10 min (i101) and 16.4% conversion after 20 min of irradiation (i201) (Fig. 1b). However, further irradiation causes the crystals to shatter and thus, it seems that 17% is the maximum conversion where the crystals can be sustained before breaking." Where do these rather precise percentage values come from? The authors have not provided any evidence for these values, or any description of how they were determined.

Line 139: "in the case of the photosalient effect"

Line 140: "However, in-situ bending of a metal-organic crystal is rare." I'm still not sure what "in situ" means here.

Line 142: "have been exposed to visible"

Line 143: "These bent crystals sustained crystallinity and diffracted nicely during SCXRD". The use of "nicely" is unscientific

Line 146: " the C=C distance decreased to 3.69 Å". I presume it's not the C=C distance that reduced, but rather the distance between neighbouring C=C moieties?

Line 182: " The cell parameters reveal that the cell volume is also increased by 8.43 Å³ (0.46%) from 1 to i51. Extensive reports show that the PS effect resulting from photodimerization is observed when there are notable changes in cell parameters. In scenarios where photodimerization causes a change in cell volume—whether it decreases or increases—the crystals can experience excessive stress, which is then relieved through mechanical motion." 0.46% is almost negligible. Are the authors trying to say that it's interesting that this value is so low, or are they saying that it is high? I can't tell.

Line 233: "couldn't" - do not use contractions

Line 253: "In one form, the crystal breaks into pieces in little to no time under UV light. On the contrary, crystals of the other form start to bend slowly, followed by flipping and swelling under the same conditions. " It's not necessarily the case that this is due to polymorphism (and I'm not convinced that these are polymorphs anyway). It could just be that the crystals behave differently because they have different mosaicities due to macroscopic changes.

Overall, I do not find the presentation of this work to be very compelling. Although it has potential to be interesting, significant additional work is required, both experimentally and in revision of the text.

Version 1:

Reviewer comments:

Reviewer #1

(Remarks to the Author)

As the authors have made the necessary and appropriate revisions, I have no objection to the manuscript being accepted

Reviewer #2

(Remarks to the Author)

The revisions made to the manuscript are insufficient for acceptance at this stage. The authors are strongly encouraged to carefully examine the referees' comments, their responses, and the modifications made in the revised manuscript. It is crucial that the comments and changes are consistent. In several instances, the authors assert that specific changes have been implemented; however, the manuscript and the accompanying electronic supplementary information (ESI) do not reflect these assertions.

Reviewer #3

(Remarks to the Author)

The authors have addressed all of my concerns and I support publication of this manuscript in its current form.

Our Response to Reviewer's Comments (COMMSCHEM-25-0053-T)

Reviewer 1

S. Khan et al. have reported the photosalient effect in a one-dimensional zinc coordination polymer. Indeed, photosalient phenomena driven by [2+2] cycloaddition in coordination polymers are rare due to the uncertainty of their aggregated structures. However, this manuscript contains several points that need revision, and therefore, a major revision is recommended.

Points for Revision

Reviewer's comment: Figure 1 is too small and difficult to see. A clearer figure should be provided.

Our Response: We are really thankful to the learned reviewer for the positive comments on our manuscript. The suggestions given are helpful for upgrading the manuscript. As suggested by the Reviewer, the size of Figure 1 has been increased. We hope the revised Figure will have better clarity.

Reviewer's comment: A figure illustrating at least the parallel alignment of C=C bonds should be included.

Our Response: We have incorporated a Figure in SI (Supplementary Fig. 2) illustrating the parallel alignment of C=C bonds.

Reviewer's comment: A graph showing the time-dependent conversion rate of the [2+2] photodimerization is necessary. It should clarify how long the irradiation takes for the conversion to reach saturation.

Our Response: As suggested by the Reviewer, we have included time-dependent conversion rate of the [2+2] photodimerization in SI (Supplementary Figs. S10 and S11).

Reviewer's comment: The paper describes single-crystal structural analysis after 390 nm light irradiation, but the method used to determine the conversion rate is unclear. Since ^1H NMR is typically used to determine conversion rates, it is important to explain how it was evaluated in this study.

Our Response: As suggested by the Reviewer, we have included time-dependent conversion rate of the [2+2] photodimerization in SI (Supplementary Figs. S10 and S11).

The photoexcited state populations are refined experimentally from the photocrystallographic data. The single-crystal X-ray diffraction data of the irradiated crystal contain atomic positions related to both the photodimerized product structure and the remaining unreacted starting structure. The atomic model for each of the states can be modelled from the diffraction data using a standard disorder model (this is done using the SHELX refinement suite by using "PART" instructions, and is a very standard approach for dealing with disordered fragments in a single crystal X-ray structure).

Each "PART" of the disordered is refined against a free variable, which essentially uses the experimental electron density to determine the precise percentage of each disorder component (here for the unreacted ground state and the photodimerized product state), which is then output from the least squares refinement of the structure model against the diffraction data.

An estimated standard deviation is produced for each occupancy value, which we have now included in the manuscript for each excited state percentage value we provide. We thank the reviewer for highlighting this

omission. Furthermore, we have provided some additional details in the supporting information document to describe the procedure for modelling the photoinduced structures from the data that explains more clearly how the photoexcited product populations are determined.

Reviewer's comment: Photos of the sample before and after the photosolient effect should be included in the main text.

Our Response: As suggested by the Reviewer, we have included the microscopic pictures of crystal before and after exhibiting the photosolient effect, as well as the bent crystal in Figure 1.

Reviewer's comment: Even if determining the structure of the complex after [2+2] photodimerization is challenging, the structure of the ligand should be accessible. Could the complex be decomposed with hydrochloric acid to extract and analyze the organic product of the [2+2] reaction?

Our Response: Thanks for the suggestion. We have performed ^1H NMR of the cyclobutane ligand by decomposing the compound using hydrochloric acid.

Reviewer's comment: Figures 3 and 4 appear to show the same data. There seems to be no need to separate them; only Fig. 4 should be retained.

Our Response: Figure 3 has been deleted and figure 4 has been retained.

Reviewer's comment: The computational results should support the experimental data. Consider whether they can be discussed in relation to the nanoindentation results.

Our Response: Further comparison of the experimental and computational values has been added.

Reviewer's comment: The meaning of the values for c11, c22, c33, c44, c55, and c66 in Figure 5 is unclear. An explanation should be provided.

Our Response: An explanation has been added to the main text.

Reviewer's comment: Figure S6 lacks a scale bar.

Our Response: Thanks. We have modified accordingly.

Reviewer's comment: The differences between Figure S8a-e are unclear, and further explanation is required.

Our Response: A description of these figures (now Fig S14) has been added to the main text.

Reviewer's comment: Examples of bending behavior induced by [2+2] cycloaddition are rare. For instance, please add the following references: Chem. Eur. J., 2024, e202401564; Chem. Mater., 2024, 36, 8338–8348; J. Mater. Chem. C, 2021, 9, 16762–16770; J. Am. Chem. Soc., 2023, 145, 6024–6028; CrystEngComm, 2025, 27, 127–137.

Our Response: As suggested by the reviewer, we have incorporated the references in the revised manuscript.

Reviewer 2

The manuscript by Khan et al. presents findings on the photosolient effects observed in a one-dimensional coordination polymer of zinc(II), an unusual phenomenon for metal-containing crystalline materials. The authors characterized the material using time-resolved photocrystallography and nanoindentation, supplementing their investigation with DFT studies. While the topic is of significant interest to a broad

audience, making it suitable for publication in the journal Communications Chemistry, the manuscript requires substantial improvements before it can be considered for acceptance.

Our Response: We are thankful to the learned reviewer for the constructive suggestions regarding our manuscript and for giving us the opportunity to revise it.

Reviewer's comment: The authors claim that there are two polymorphic forms of the zinc-based 1-D polymer. If this is indeed the case, the preparation details for both forms should be clearly articulated, emphasizing the differences in their preparation methods. The information provided regarding the preparation of form II is insufficiently detailed; specifically, it is unclear whether both forms were stored in the dark, how long the crystals remained in their mother liquors, and what procedures were followed after the crystals were isolated—were they also kept in the dark, for instance?

Additionally, it would be ideal to demonstrate phase purity by presenting PXRD patterns, along with their comparisons to the respective patterns calculated from the crystal structures and/or their mutual overlaps. Furthermore, a systematic labeling of the two forms in the manuscript would be necessary.

Our Response: The reaction is performed by slow diffusion process and kept in dark. Within a week, needle crystals (compound **1**) began to form, and when the reaction mixture is kept for longer time, the bent crystals (compound **1b**) appeared in the same reaction system. Both types of crystal are kept in the mother liquor in the dark until the SCXRD or other characterizations are performed. This has been clarified in the text.

Also, as suggested we have performed the PXRD of both the crystals to check the phase purity.

We now have mentioned needle crystals as **1** and bent crystals as **1b** throughout the manuscript.

Reviewer's comment: In the abstract, the authors claim that during 'photoinduced transformation', the crystals exhibited diverse photomechanical movements, such as jumping, bending, flipping, splitting, and swelling in response to UV light.' However, only bending and splitting/explosions were described in the MS.

Our Response: We would like to thank the learned reviewer for careful observation. We have revised the abstract and kept only jumping, bending and splitting/explosions as described in the manuscript.

Reviewer's comment: The nanoindentation was performed on (0-11) crystal face, but it is unclear why this specific face was selected for analysis. What other faces were developed on the crystals of the two forms? Additionally, what is the relationship between the crystal structures and the crystal morphology (i.e., crystal faces) of these two forms? Similarly, regarding the DFT studies, it is not clear why the authors chose to focus on the strain induced in the c direction; it should be mentioned.

Please pay attention to the distinctions in labeling crystallographic directions and crystal faces.

Our Response: From the face indexing, the most prominent/ major face was (0-11), where indentation was performed. Other faces are shown in the inset of Figure 6b, which are minor faces. Nanoindentation could only be performed on the major face (0-11), as other faces were inaccessible due to uneven surfaces and experimental challenges. We have also provided a schematic of the crystal and packing diagram in the SI. A clarification on the choice of c axis has been added to the methodology.

Reviewer's comment: The differences in moduli (Shear, Bulk, Young's) before, during and after bending in relation to 'elastic stiffness constants' (c_{11} , c_{22} ..., c_{66}) were discussed, and the behaviors were further related with the reduced/increased brittleness/flexibility. It would be beneficial to provide a more detailed explanation (in ESI) why the authors selected these specific matrix elements and how their changes relate to the brittleness and flexibility of crystals. Additionally, the authors mentioned visualizing the calculated stiffness constants by the ELATE/VESTA program Including 3D presentations and/or 2D sections in the main text or ESI would be advantageous.

Our Response: More details on these matrix elements and their physical meaning and why they are discussed have been added to the main text. ELATE-generated 2D and 3D representation have been added in the SI.

Reviewer's comment: While Scheme 1 is a valuable addition, a more comprehensive scheme that includes additional details would be more appropriate. Such a scheme would aid readers in understanding the complex behavior of the CP-based material.

Our Response: Thanks for the suggestion. We have revised Scheme 1.

Reviewer's comment: Page 11, ln 228; refs [42-44]; please check the reliability of the citation.

Our Response: We thank the referee for pointing out this. We have now corrected this revised manuscript.

Reviewer 3

Reviewer's comment: In my view this work is way too premature for publication. The results may be of potential interest to the crystal engineering community, but (1) the manuscript is poorly written, and (2) insufficient supporting evidence is provided for the discussion and conclusions. The presentation of the work is sloppy and the results don't stand out as being unusual for crystals that undergo [2+2] cycloaddition reactions. How reproducible are the nanoindentation results?

Our Response: We are really thankful to the learned reviewer for the positive comments on our results and for giving us the opportunity to revise. The comments are really helpful to upgrade the quality of manuscript.

In Figure 6e, f, histogram plots for the distribution of the elastic modulus and hardness are given. The nanoindentation was done on multiple single crystals, and the average of the values are presented in text and Table 1. Moreover, Figure S13b shows the scatter plot. All of these data points/figures from multiple indentations are a routine representation for nanoindentation study, indicating the reproducibility of the nanoindentation.

Reviewer's comment: In addition to the above, the following needs to be addressed if the authors are planning to submit this work again:

Our Response: We are thankful to the learned reviewer for the constructive suggestions regarding our manuscript and for giving us the opportunity to revise it.

Reviewer's comment: The formula $[Zn(DCTP)_2(rctt-4-pncb)]_n$ is incorrect everywhere in the manuscript. It should be $[Zn(DCTP)(rctt-4-pncb)]_n$ {note, there is only one DCTP per Zn ion}

Our Response: Many thanks for pointing this out. We have now modified the formula in the revised manuscript following the reviewer's suggestion.

Reviewer's comment: Inclusion of a chemical scheme showing the formula unit would be useful to the reader.

Our Response: We have incorporated a scheme (Supplementary Scheme 1) in the SI as suggested. Thanks.

Reviewer's comment: Line 47: These materials, after light adsorption, generate and accumulate strain ... "adsorption" should be "absorption"

Our Response: Corrected. Thanks

Reviewer's comment: Line 49: "di-arylethylenes" should be "diarylethylenes"

Our Response: Corrected. Thanks

Reviewer's comment: Line 50: insert "of" between "kind" and "structural"

Our Response: Corrected. Thanks

Reviewer's comment: Line 55: "energetic" should be "energetics"

Our Response: Corrected. Thanks

Reviewer's comment: Line 55: "resulting in motion" - insert "in"

Our Response: Corrected. Thanks

Reviewer's comment: Line 55/56: change to "observed macroscopically"

Our Response: Corrected. Thanks

Reviewer's comment: Line 57: "crystals violently explode and shatter into pieces" - things that explode generally shatter into pieces, so this is a tautology

Our Response: Corrected. Thanks

Reviewer's comment: Line 61: change "tools" to "tool"

Our Response: Corrected. Thanks

Reviewer's comment: Line 71: "Of the external stimuli, actuation by light comes with advantages for control over the macroscopic shape in comparison to mechanical or temperature-driven most notably, the possibility for remote control and precise input of energy" very poor sentence structure.

Our Response: We have revised the sentence as suggested. Thanks.

Reviewer's comment: Line 73: "In this regard, in-situ photomechanical bending is even more intriguing since no input of external energy or harmful UV light is needed". If bending is due to irradiation, then how can there be no input of external energy?? The inclusion of "harmful" seems spurious because this doesn't need to be an issue for the manipulation of materials.

Our Response: We have deleted the word "harmful". Thanks.

Reviewer's comment: Line 81: "formulae" should be "formula"

Our Response: Corrected. Thanks

Reviewer's comment: Line 85: "Unexpectedly, bent shaped single crystals (1b) were also obtained in-situ without applying any external stimuli". Firstly, it's not clear what "in situ" is referring to. Do the authors mean in situ in the vial, or in situ in the X-ray beam? Either way, there are external stimuli that cannot be ruled out, including temperature, ambient light and (possible) X-ray photons.

Our Response: Thanks for the comment. 'In situ' refers to the crystal bending observed directly in the reaction vial without intentional external stimuli. This has been clarified in the text.

Reviewer's comment: Line 86: "We also observed two polymorphs with different photomechanical properties." On what basis have the authors decided that 1 and 1b are polymorphs. This is never really discussed. Besides, the unit cell parameters, are similar and the space groups are the same. The packing arrangements are also sufficiently similar to suggest that these are not polymorphs.

Our Response: We thank the Reviewer for pointing this out. We agree that these compounds cannot be considered as polymorphs. We have revised the manuscript accordingly.

Reviewer's comment: Line 87: "wherein crystal" - insert "the"

Our Response: Corrected. Thanks

Reviewer's comment: Line 89: "flipping and swelling and this form also grow in-situ which may be due to the slight absorption of visible light". Change "grow" to "grows". Again, what does "in situ" refer to? The vial or the X-ray beam?

Our Response: Corrected. Here, 'in situ' refers to the crystal bending observed directly in the reaction vial without intentional external stimuli. Thanks.

Reviewer's comment: Line 107: "forms a strong hydrogen bond" - insert "a" as shown

Our Response: Corrected. Thanks

Reviewer's comment: Line 109: "generate a 1D CP"

Our Response: Corrected. Thanks

Reviewer's comment: Line 110: "exposed at both sides of the chain"

Our Response: Corrected. Thanks

Reviewer's comment: Line 116: " In addition, in situ bent crystals (1b) are obtained by keeping the reaction mixture for longer time." Omit "time"; again, what does in situ mean?

Our Response: Corrected. Here, 'in situ' refers to the crystal bending observed directly in the reaction vial without intentional external stimuli. Thanks

Reviewer's comment: Line 132: "A photocrystallographic study of a [2+2] cycloaddition reaction involves using photocrystallography to investigate the [2+2] cycloaddition reaction". This sentence contains no useful information, and states the obvious twice.

Our Response: Corrected. Thanks

Reviewer's comment: Line 127: "The reason behind choosing the certain wavelength, in the solid-state, it is typical to choose a wavelength in the tail of the absorption band." Very poor sentence construction

Our Response: We have revised the sentence as suggested. Thanks.

Reviewer's comment: Line 132: "On irradiation at 390 nm, it resulted in 12.4% conversion to the photodimerized product after 5 min (i51), 15.1% conversion after 10 min (i101) and 16.4% conversion after 20 min of irradiation (i201) (Fig. 1b). However, further irradiation causes the crystals to shatter and thus, it seems that 17% is the maximum conversion where the crystals can be sustained before breaking." Where do these rather precise percentage values come from? The authors have not provided any evidence for these values, or any description of how they were determined.

Our Response: The photoexcited state populations are refined experimentally from the photocrystallographic data. The single-crystal X-ray diffraction data of the irradiated crystal contain atomic positions related to both the photodimerized product structure and the remaining unreacted starting structure. The atomic model for each of the states can be modelled from the diffraction data using a standard disorder model (this is done using the SHELX refinement suite by using "PART" instructions, and is a very standard approach for dealing with disordered fragments in a single crystal X-ray structure).

Each "PART" of the disordered is refined against a free variable, which essentially uses the experimental electron density to determine the precise percentage of each disorder component (here for the unreacted ground state and the photodimerized product state), which is then output from the least squares refinement of the structure model against the diffraction data.

An estimated standard deviation is produced for each occupancy value, which we have now included in the manuscript for each excited state percentage value we provide. We thank the reviewer for highlighting this omission. Furthermore, we have provided some additional details in the supporting information document to describe the procedure for modelling the photoinduced structures from the data that explains more clearly how the photoexcited product populations are determined.

Reviewer's comment: Line 139: "in the case of the photosolient effect"

Our Response: Corrected. Thanks

Reviewer's comment: Line 140: "However, in-situ bending of a metal-organic crystal is rare." I'm still not sure what "in situ" means here.

Our Response: Here, 'in situ' refers to the crystal bending observed directly in the reaction vial without intentional external stimuli. Thanks

Reviewer's comment: Line 142: "have been exposed to visible"

Our Response: Corrected. Thanks

Reviewer's comment: Line 143: "These bent crystals sustained crystallinity and diffracted nicely during SCXRD". The use of "nicely" is unscientific

Our Response: We thank the reviewer for pointing this out. We have omitted nicely and modified the sentence accordingly.

Reviewer's comment: Line 146: " the C=C distance decreased to 3.69 Å". I presume it's not the C=C distance that reduced, but rather the distance between neighbouring C=C moieties?

Our Response: We thank the reviewer for pointing this out. We have omitted nicely and modified the sentence accordingly.

Reviewer's comment: Line 182: " The cell parameters reveal that the cell volume is also increased by 8.43 Å³ (0.46%) from 1 to i51. Extensive reports show that the PS effect resulting from photodimerization is observed when there are notable changes in cell parameters. In scenarios where photodimerization causes a change in cell volume—whether it decreases or increases—the crystals can experience excessive stress, which is then relieved through mechanical motion." 0.46% is almost negligible. Are the authors trying to say that it's interesting that this value is so low, or are they saying that it is high? I can't tell.

Our Response: Literature reports suggest that a modest increase in cell volume (1.0–2.0%) typically indicates the presence of the PS effect. In this study, a ~0.5% increase in cell volume is observed, despite only ~17% photodimerization. Although this change is relatively small, it implies that more substantial structural changes could be expected if full (100%) dimerization were achieved. This has been clarified in the text.

Reviewer's comment: Line 233: "couldn't" - do not use contractions

Our Response: Corrected. Thanks

Reviewer's comment: Line 253: "In one form, the crystal breaks into pieces in little to no time under UV light. On the contrary, crystals of the other form start to bend slowly, followed by flipping and swelling under the same conditions. " It's not necessarily the case that this is due to polymorphism (and I'm not convinced that these are polymorphs anyway). It could just be that the crystals behave differently because they have different mosaicities due to macroscopic changes.

Our Response: We thank the Reviewer for this observation and agree that the compounds should not be classified as polymorphs. The manuscript has been revised accordingly. As stated by the Reviewer, the observed differences in the photomechanical effect are attributed to variations in crystal morphology (shape and size), consistent with previous reports. This point has been added to the main text.

Our Response to Reviewer's Comments (COMMSCHEM-25-0053A)

Reviewer #1 (Remarks to the Author):

As the authors have made the necessary and appropriate revisions, I have no objection to the manuscript being accepted

Our Response: We are grateful to the learned reviewer for supporting and positive evaluation of our manuscript.

Reviewer #2 (Remarks to the Author):

The revisions made to the manuscript are insufficient for acceptance at this stage. The authors are strongly encouraged to carefully examine the referees' comments, their responses, and the modifications made in the revised manuscript. It is crucial that the comments and changes are consistent. In several instances, the authors assert that specific changes have been implemented; however, the manuscript and the accompanying electronic supplementary information (ESI) do not reflect these assertions.

Our Response: We sincerely thank the reviewer for their careful reading and for highlighting the need for consistency. In response, we have conducted a thorough cross-check of our revised manuscript and supplementary information to ensure that all previously claimed changes are fully and accurately implemented. We confirm the following:

1. **Polymorph Clarification:** The terms “polymorphs” have been removed. The two forms are now consistently referred to as **1** and **1b**, and PXRD patterns for both are included (Supplementary Figs. S3 and S4).
2. **Photomechanical Behavior:** The abstract has been revised to reflect only the observed effects (jumping, bending, and splitting), consistent with the main text and figures.
3. **Nanoindentation Face Selection:** The rationale for selecting the (0-11) face is now clearly stated in the main text, and schematic of the crystal faces are now provided (Supplementary Fig. S14-15).
4. **DFT Strain Direction:** The choice of strain direction along the c-axis is now explained in the Computational Methods section. It was mistakenly not highlighted in the revised version.
5. **Elastic Constants (c11–c66):** These are discussed in the main text (Fig. 4b), with full stiffness matrices (Tables S3–S7) and ELATE visualizations (Figs. S18–S20) included in the SI.
6. **Scheme 1:** Revised to better illustrate the photomechanical behavior and computational workflow.
7. **Reference Corrections:** The citations flagged by the reviewer (e.g., Refs. 42–44) have been corrected.

We hope this clarifies that all reviewer suggestions have been fully implemented and are now reflected in the revised manuscript and SI.

Reviewer #3 (Remarks to the Author):

The authors have addressed all of my concerns and I support publication of this manuscript in its current form

Our Response: We are grateful to the learned reviewer for supporting and positive evaluation of our manuscript.

Our Response EDITORIAL REQUESTS (COMMSCHEM-25-0053B)

* Your manuscript should comply with our policies and format requirements, detailed in our style and formatting guide (<https://www.nature.com/documents/commsj-phys-style-formatting-guide-accept.pdf>).

Our Response: We have now followed policies and format requirements. Thanks

* Please edit your manuscript according to the editorial requests in the attached table, and outline revisions made in the right hand column. If you have any questions or concerns about any of our requests, please do not hesitate to contact me. It is important that each request be addressed in order to avoid delays in accepting your manuscript. Please upload the completed table with your manuscript files as a Related Manuscript file.

Our Response: We have now followed all the guidelines. Thanks